# Using Measles Outbreaks to Identify Under-Resourced Health Systems in Low- and Middle-Income Countries: A Predictive Model

**DOI:** 10.3390/vaccines13040367

**Published:** 2025-03-30

**Authors:** Gabrielle P. D. MacKechnie, Milena Dalton, Dominic Delport, Stefanie Vaccher

**Affiliations:** 1Burnet Institute, Melbourne, VIC 3004, Australia; 2Melbourne School of Population and Global Health, University of Melbourne, Melbourne, VIC 3010, Australia; 3Department of Epidemiology and Preventive Medicine, Monash University, Melbourne, VIC 3004, Australia

**Keywords:** measles, disease outbreaks, epidemics, health systems, healthcare, health services

## Abstract

Background/Objectives: Measles is a vaccine-preventable disease with a high level of transmissibility. Outbreaks of measles continue globally, with gaps in healthcare and immunisation resulting in pockets of susceptible individuals. Measles outbreaks have been proposed as a “canary in the coal mine” of under-resourced health systems, uncovering broader system weaknesses. We aim to understand whether under-resourced health systems are associated with increased odds of large measles outbreaks in low- and middle-income countries (LMICs). Methods: We used an ecological study design to identify measles outbreaks that occurred in LMICs between 2010 and 2020. Health systems were represented using a set of health system indicators for the corresponding outbreak country, guided by the World Health Organization’s building blocks of health systems framework. These indicators were: the proportion of births delivered in a health facility, the number of nurses and midwives per 10,000 population, and domestic general government health expenditure per capita in USD. We analysed the associations using a predictive model and assessed the accuracy of this model. Results: The analysis included 78 outbreaks. We found an absence of any association between the included health system indicators and large measles outbreaks. When testing predictive accuracy, the model obtained a Brier score of 0.21, which indicates that the model is not informative in predicting large measles outbreaks. We found that missing data did not affect the results of the model. Conclusions: Large measles outbreaks were not able to be used to identify under-resourced health systems in LMICs. However, further research is required to understand whether this association may exist when taking other factors, including smaller outbreaks, into account.

## 1. Introduction

Measles is a vaccine-preventable disease; however, outbreaks continue to occur. The disease mainly affects children, and complications of infection with this airborne virus can be severe and include pneumonia, encephalitis, ear infection, neurological complications, and death [1]. Since the introduction of the measles vaccine in 1963, there has been a significant reduction in the number of cases and deaths associated with the disease [2,3]. Measles vaccination has averted 56 million deaths worldwide between 2000 and 2021 [3]. Despite this, measles remains a public health burden. In 2021, there was an estimated incidence of 9 million cases and 128,000 measles-related deaths globally [4]. The number of large and disruptive measles outbreaks (those with an incidence of at least 20 cases per million population over a period of 12 months) has increased consistently since January 2022 [5,6]. The burden of measles varies substantially across regions, with particularly poor outcomes in low- and middle-income countries (LMICs) [7]. In fact, 99.7% of all reported measles cases in 2022 occurred in LMICs [8].

The burden of measles is driven by its high transmissibility, with a basic reproduction number (R0) estimated between 12 and 18 [9]. Therefore, 95% vaccination coverage with two doses of a measles-containing vaccine is necessary to achieve herd immunity [10]. The accumulation of susceptible individuals can lead to outbreaks of measles, particularly among those without complete vaccination [11]. Even when high national vaccination coverage is reported, marginalised, vulnerable, or conflict-affected communities may be missed by vaccination programmes, resulting in variation at sub-national levels [12]. This variation in immunisation coverage is driven by health inequities, arising from differences in the distribution of health system services [13]. Institutionalised disparities in privilege between groups (e.g., racism, homophobia), known as systemic inequities, can influence the allocation of health services, as well as social and environmental infrastructure [13]. The Gavi 2021–2025 strategy [14] aims to address these health inequities by focusing immunisation programmes on missed communities and zero-dose children (children who have not received the first dose of the diphtheria–tetanus–pertussis vaccine) [15]. In LMICs, challenges to achieving high, uniform vaccination coverage are exacerbated by a decreased uptake in routine childhood vaccines following the COVID-19 pandemic and the effects of previous parasite infections on vaccine-induced immune responses [16,17]. In under-resourced settings, where health systems lack sufficient infrastructure to deliver effective vaccination programmes and control outbreaks of the disease, measles can become a considerable public health burden [18]. Countries or regions with sub-optimal measles immunisation coverage have a higher likelihood of measles outbreaks, posing the risk of increased morbidity and mortality, as well as severe social and economic outcomes [3,19].

Measles is often described as a ‘canary in the coal mine’ of under-resourced health systems [12,20,21]. The World Health Organization (WHO) Immunisation Agenda 2030 [12] and the Measles and Rubella Strategic Framework 2021–2030 [20] establish that measles outbreaks reveal vulnerabilities in the healthcare system due to the high transmissibility of the disease. Similarly, John [21] reported that measles may uncover gaps in immunity, which, in turn, indicate underlying problems in health infrastructure. An article from WHO [18] discusses the overwhelmed state of health systems following the COVID-19 pandemic, creating a ‘perfect storm’ for disruptive measles outbreaks, which may serve as a warning for the emergence of other vaccine-preventable infectious diseases. Populations with fragmented health systems often lack high-quality data on immunisation coverage and other health system performance indicators, hindering effective prioritisation of resources [22,23,24]. Applying measles outbreaks as a tracer of health system weaknesses could address this issue.

Robust health systems, including strong surveillance systems, are crucial to preventing and responding to outbreaks of vaccine-preventable diseases [12,25]. Accessible health services, a strong health workforce, and efficient communication within the health system encourage high immunisation coverage [25,26]. Surveillance plays a critical role in early detection and response, therefore limiting the size of emerging outbreaks [25,27]. The WHO describes the core components of the health system using six distinct building blocks: health service delivery, the health workforce, access to essential medicines, health information systems, health financing, and leadership and governance [28]. The building blocks framework has been widely used in the health system strengthening literature [29,30,31], while also serving as a basis for healthcare evaluation toolkits [32,33]. In this analysis, we examine these fundamental aspects of the health system to unveil the underlying gaps that hinder outbreak prevention and response. The health system can also be described on a national, community or individual level. Although under-resourced health systems exist in both high-income countries and LMICs, the latter are disproportionately affected, with an estimated 8 million deaths annually due to conditions that should be treatable in a high-quality health system [34]. Therefore, this analysis focuses on health systems in LMICs.

Despite reports of measles as an early indicator of insufficient health systems, to our knowledge, there has not been any formal quantitative analysis on the topic. An understanding of this topic may reveal the most vulnerable aspects of the health system, allowing public health coordinators and policymakers to identify priorities in health planning. Consequently, countries may be able to prevent future measles outbreaks and other public health disasters while strengthening the health system more broadly.

This study aims to determine whether under-resourced health systems are associated with increased odds of large measles outbreaks in LMICs.

## 2. Materials and Methods

Our analysis used an ecological study design to understand, at a national level, whether under-resourced health systems are associated with increased odds of large measles outbreaks in LMICs. Using publicly available secondary data, we fitted a predictive model to estimate this association. Ethical approval was not required, as this project only used publicly available, non-identifiable data measured at the group level.

### 2.1. Directed Acyclic Graph

First, we developed a directed acyclic graph (DAG) using the DAGitty version 3.1 software [35] in consultation with the literature and expert knowledge to identify potential sources of selection or information bias.

### 2.2. Data Collection

#### 2.2.1. Measles Outbreaks

We defined a measles outbreak as two or more laboratory-confirmed measles cases with rash onset within 7–21 days of each other, in addition to being epidemiologically and/or virologically linked [36]. A country was considered an LMIC if it was listed by the World Bank as a low-, lower-middle, or upper-middle-income country in the calendar year when the outbreak began [37]. Using these criteria, we searched the websites ReliefWeb [38] and WHO Disease Outbreak News [39] for past measles outbreaks in LMICs and removed any duplicate outbreaks. Each measles outbreak corresponded to one observation in our dataset. We included outbreaks in LMICs with a start date between 1 January 2010 and 11 March 2020 (the date the COVID-19 pandemic was declared), including ongoing outbreaks [40]. During the COVID-19 pandemic, measles surveillance systems were disrupted, so we limited the outbreak date range to reduce potential bias arising from under-reporting of measles cases [41,42]. For ongoing outbreaks, we used the most recent outbreak reports as of 12 September 2023 to ensure case numbers were as accurate as possible. When multiple outbreaks occurred in the same country across non-overlapping time periods, all outbreaks were included in the data.

Subsequently, we collected data on the number of cumulative cases for each identified outbreak, using WHO regional epidemiological reports, and supplemented by ReliefWeb and Disease Outbreak News reports [38,39,43,44,45,46,47,48]. When necessary, we collated case numbers over multiple (non-overlapping) reporting periods, and included all laboratory-confirmed, clinically confirmed, epidemiologically linked and suspected cases. The data is provided in Appendix A.

#### 2.2.2. Health Systems

We represented the capacity of health systems using three health system indicators aggregated at the national level. To select indicators, we devised suitable indicators corresponding to each health system building block, guided by the WHO building blocks of health systems [28]. To minimise bias, we excluded building blocks with limited indicator data across a large number of LMICs (access to essential medicines, health information systems, and leadership and governance) [28]. The final indicators were the proportion of births delivered in a health facility (corresponding to health service delivery), the number of nurses and midwives per 10,000 population (health workforce), and domestic general government health expenditure per capita in USD (health financing) [28].

For each identified measles outbreak, we collected data from the WHO Global Health Observatory [49,50,51] regarding the above indicators for the outbreak country. To ensure temporality, we collected health system indicator data measured in the calendar year prior to the start of the measles outbreak where available; otherwise, we used the most recent year for which data were available before the outbreak.

#### 2.2.3. Gini Coefficient

We used the Gini coefficient as a proxy for systemic inequity. The Gini coefficient quantifies income inequality, in addition to other forms of inequality, and can be compared across countries [52]. For each observation, we collected Gini coefficient data from the World Bank [53] for the most recent calendar year prior to the start of the outbreak in the corresponding country.

#### 2.2.4. Proportion of Refugees and Asylum Seekers

To account for the burden of measles in conflict-affected populations, we incorporated a variable representing the number of refugees and asylum seekers as a percentage of each country’s population. We collected data from the UNHCR Refugee Data Finder [54] for the most recent calendar year prior to the start of the outbreak in the corresponding host country.

### 2.3. Data Processing

To prepare the data for analysis, we clustered the observations by each outbreak country, using the mean of each variable as a summary statistic in each cluster. Clustering methods were used to account for the possibility of non-independence of observations for outbreaks that occurred in the same country, resulting from similarities in their health system characteristics. We then created a binary variable for large measles outbreaks, derived from the cumulative case data. The WHO defines a large and disruptive measles outbreak as having an incidence of at least 20 laboratory-confirmed, clinically confirmed, or epidemiologically linked cases per million population over 12 months [5,55]. Investigations of past measles outbreaks in LMICs have found that between 16% and 25% of suspected cases are laboratory-confirmed, clinically confirmed, or epidemiologically linked [56,57,58]. As our analysis includes suspected, laboratory-confirmed, clinically confirmed, and epidemiologically linked cases, we defined a large outbreak as having an incidence equal to or greater than 100 cases per million population over a 12-month period, as we expect approximately 20 of these cases to be laboratory-confirmed, clinically confirmed, or epidemiologically linked. To calculate incidence, we used population size estimates from the World Bank [59] and defined outbreak duration as the total number of years between the first and the last years of the outbreak, inclusive. Since our analysis compares outbreaks across various countries, we did not use country-specific thresholds for the definition of a large outbreak. We conducted a sensitivity analysis for different definitions: incidence greater than or equal to 20 cases, 50 cases, 150 cases, and 200 cases per million population over the course of 12 months. We also conducted a sensitivity analysis based on the WHO definition of a large measles outbreak. Since the data sources used in the main analysis lacked consistent data on laboratory-confirmed, clinically confirmed, and epidemiologically linked cases, we instead used data from the WHO Global Health Observatory [54] for this sensitivity analysis. While not outbreak-specific, these data report the number of confirmed and epidemiologically linked measles cases for each country and year.

We did not scale the health system indicator variables, since this study is not focused on the interpretation of individual regression coefficients, but rather, the broader association between under-resourced health systems and large measles outbreaks.

### 2.4. Analysis Methods

Stata version 17 (StataCorp, College Station, TX, USA) was used for analysis. In the initial analysis, we calculated measures of central tendency and spread for all variables. The dataset was randomly allocated into training and testing datasets, using an 80:20 training/testing split. The training dataset was used to fit a regression model, whereas the testing dataset was used to assess the predictive performance of the model.

#### 2.4.1. Regression Model Fitting and Testing

We constructed a multivariable logistic regression model, with health system indicators as inputs and large measles outbreaks as the output. We conducted diagnostics for linearity, goodness of fit, and influential observations to ensure our model did not violate the assumptions of regression. We removed any influential observations by assessing standardised residuals, deviance, and leverage. Additionally, we tested for collinearity between the health system indicator variables.

We used the regression output to predict the odds of large measles outbreaks for observations in the testing dataset. Subsequently, we assessed the external validity of the model by comparing the predicted odds against the observed outcomes, using a plot and the Brier score [60]. A model that perfectly predicts the outcome has a Brier score of 0, whereas a model with perfect inaccuracy has a Brier score of 1 [60]. A model that gives a 50% chance of the outcome (and is therefore non-informative) gives a Brier score of 0.25 [60].

#### 2.4.2. Sensitivity Analyses

To test for effect modification, we constructed the logistic regression model with an interaction term between the Gini coefficient variable and each of the input variables, then assessed the output of the interaction terms. Additionally, we conducted a sensitivity analysis incorporating the proportion of refugees and asylum seekers in each country as a covariate. Furthermore, we tested whether missing data influenced the results. We handled all missing values using multiple imputation and fitted and tested the logistic regression model using the imputed dataset.

## 3. Results

### 3.1. Directed Acyclic Graph

The DAG for this study is shown in Figure 1. In the DAG, we included measles immunisation coverage, given the critical role of the health system in reaching high immunisation coverage and the impact of immunisation gaps on measles outbreaks [11,12]. Health inequities were also incorporated into the DAG, acknowledging that such inequities stem from the allocation of health system resources, leading to variation in immunisation coverage and an increased likelihood of measles infection [12,13]. Systemic inequities influence the distribution of health services, both directly, through institutionalised attitudes towards groups of people, and indirectly, through differences in social and environmental conditions [13]. We also considered the measurement of the health system and large measles outbreaks, through both health system reporting and surveillance systems. In the analysis, we did not condition on surveillance systems, as this is a collider variable. Similarly, we did not adjust for measles immunisation coverage nor health inequities, as both these variables are mediators, with measles immunisation coverage also being a collider variable.

### 3.2. Descriptive Statistics

In total, we identified 78 measles outbreaks that occurred in LMICs over the study period, corresponding to 55 clusters after collapsing the data by outbreak country. Across the clusters, 28 (52.8%) were classified as large; two outbreaks were not classified due to missing cumulative case data. Among countries with large measles outbreaks, the median proportion of births delivered in a health facility was 74.9%, compared to 88.6% for non-large measles outbreaks (Table 1). The median number of nurses and midwives per 10,000 population was also lower for large outbreaks (11.0) compared to non-large outbreaks (18.3), as was health expenditure.

### 3.3. Regression Analysis

Forty-four observations were randomly allocated to the training dataset; however, only thirty-eight of these were used to fit the logistic regression model due to missing data. Eleven observations were randomly allocated to the testing dataset, all of which were used to predict the odds of the outcome. The likelihood ratio test for effect modification due to systemic inequities found weak evidence against the null hypothesis of no effect modification (*p*-value = 0.25). Therefore, the Gini coefficient was not included in the final model.

From the output of the multivariable regression model (Table 2), all coefficients for the health system indicator input variables are close to the null value of one. The regression model returned a pseudo-R^2^ estimate of 8.5%, meaning that only 8.5% of the variance in the large outbreaks variable could be predicted by the health system indicator inputs. We interpret this with caution as it is a pseudo-estimate.

After making predictions for the testing dataset, the model obtained a Brier score of 0.210, indicating that the model does not effectively predict large measles outbreaks and, therefore, lacks external validity. Figure 2 shows the predicted odds of large measles outbreaks for each observation in the testing dataset, grouped by the observed outcome (large outbreak or non-large outbreak). Outbreaks that were observed to be large had a higher median predicted odds of being large compared to outbreaks that were observed to be non-large. Despite this result, observed non-large outbreaks still had a median predicted odds close to the null value of one. Consequently, the health system indicators in the model could not effectively predict large measles outbreaks.

### 3.4. Sensitivity Analyses

When fitting regression models for different definitions of a large measles outbreak, it was observed that all models produce similar model coefficients and validation results to the baseline definition. These results confirm that the selected definition of a large measles outbreak did not impact the results of the model. Similarly, adding the proportion of refugees and asylum seekers in each country as a covariate did not alter the findings of the analysis. Results of the sensitivity analyses can be found in Appendix B.

### 3.5. Missing Data

When fitting the regression model on imputed data, we obtained similar model coefficients and validation results to the output of the non-imputed data, indicating that the model does not effectively predict the outcome. We expect that missing data did not affect the results.

## 4. Discussion

### 4.1. Key Findings

We found no association between the selected health system indicators and the odds of a large measles outbreak. The selected indicators could not effectively predict large measles outbreaks. Our results show that measles outbreaks may not be an effective indicator of under-resourced health systems in LMICs.

Considering these results, it is important to review our characterisation of health systems, for which we used only three indicators as inputs to the model. When selecting indicators, we did not identify any indicators that sufficiently characterised the health information systems or leadership and governance building blocks. After reviewing potential data sources, we found a lack of complete data for the Average availability of 14 selected essential medicines in public health facilities, which was our selected indicator for the access to essential medicines building block. Therefore, this study only considered indicators for health service delivery, health workforce, and health financing building blocks [28]. The importance of these particular health system components in the control of other infectious diseases has been emphasised in the literature. A systematic review by Shoman et al. [61] found that the health workforce (including nurses and physicians) is paramount to Ebola outbreak control, as is health service delivery. Importantly, this review noted that successful health service delivery depends on the success of the other building blocks; therefore, by including an indicator for service delivery, our model has encapsulated the health system [61]. Such a relationship between the building blocks also suggests that including additional building block indicators or a surveillance system measure in our model may have led to collinearity; therefore, the selection of only three indicators is justified. The reliance of surveillance systems on the health system, coupled with their role in measles outbreak reporting, may have introduced collider bias had they been incorporated into our model. However, our selection of indicators may have obscured some aspects of the health system from the predictive model, attenuating the association between under-resourced health systems and measles outbreaks.

Another possible explanation for why our findings did not demonstrate an association is that our model did not incorporate the effect of fragile and conflict-affected settings on the health system. The presence of conflict may disrupt health services and access to healthcare, and measles outbreaks are frequently reported in refugee camps and conflict-affected populations [62,63,64]. Relating health systems to displaced populations presents challenges due to migration and limited healthcare access [63]. Although our sensitivity analysis included a measure for refugees and asylum seekers, it may not capture the broader impact of conflict on the health system. While not specifically examined in this study, further research is warranted to determine appropriate health system indicators for displaced populations. Conflict and insecurity are often localised to a sub-national level, which poses challenges in incorporating meaningful measures of conflict into our analysis, conducted at the national level [62]. Failing to adjust for conflict may introduce potential confounding, although given that we are not making causal inferences on the association, this effect is of less concern.

This project used an ecological study design with health system indicators measured at the national level, which may mask associations occurring at the sub-national level. Under-resourced health systems may lead to varying measles vaccination coverage between communities and regions, resulting in increased measles susceptibility in some areas [64]. Certain marginalised groups often face physical, cultural, and social barriers to accessing healthcare, in addition to unequal distribution of health services and resources by the health system [64]. Missed communities have become a key focus in the current approach to immunisation delivery, as they are a priority population in the Gavi 2021–2025 strategy [14]. This issue is especially pertinent to the high transmissibility of measles, as outbreaks occur in pockets of susceptible individuals [11]. Further research is required to determine whether under-resourced health systems are associated with measles outbreaks on a sub-national or individual level.

We focused on comparing large measles outbreaks to non-large measles outbreaks. In addition to the public health significance of large measles outbreaks, this comparison provided a more robust analysis than investigating outbreaks of any size against the absence of outbreaks. Consequently, our study has only somewhat ascertained whether measles outbreaks can be used to identify under-resourced health systems in LMICs. The components of weak health systems may still be significant in determining the presence or absence of any measles outbreak, which was not explored in our analysis. This association is largely dependent on outbreak prevention, particularly the presence of herd immunity, achieved through 95% vaccination coverage with two doses of a measles-containing vaccine [10]. Vaccine effectiveness is critical to achieving herd immunity, requiring strong health systems to ensure that an immune response is induced [16]. This is important, as variation in measles immunisation coverage can lead to measles outbreaks [11]. Conversely, the size of existing measles outbreaks depends on both prevention and control measures, such as reactive vaccination campaigns [12,65,66]. Once a measles outbreak is established, an effective and timely response is critical to circumventing a large outbreak [65]. Consequently, the capacity of health systems to implement preventive measures and avert measles outbreaks may have been attenuated in this analysis. Therefore, the concept of measles as a proxy for under-resourced health systems, as put forward in the Immunization Agenda 2030 [12] and other public health papers [20,21], may still be relevant to the application of health system strengthening.

### 4.2. Strengths and Limitations

In this study, there were challenges in ensuring that all measles cases stated in the epidemiological reports belonged to the same measles outbreak, resulting in potential misclassification of non-large outbreaks as large outbreaks, as well as different case definitions used across various countries [67]. However, by capturing all suspected, epidemiologically linked, clinically confirmed, and laboratory-confirmed cases in the measles case data, we reduced information bias resulting from the possibility of weaker surveillance systems and diagnostic capacities in countries with under-resourced health systems. As we restricted the data to outbreaks that started before the COVID-19 pandemic, our findings are not generalisable to post-COVID-19 settings, given the effect that the COVID-19 pandemic has had on health systems and vaccination programmes [17]. Despite this, our study produced evidence on a topic of global interest that, to our knowledge, has not yet been investigated empirically.

### 4.3. Implications

Measles outbreaks have been put forward as a “canary in the coal mine” for broader health system weaknesses, suggesting that they may be used to identify health systems most vulnerable to infectious disease outbreaks and other health issues [12,20,21]. However, our analysis has found that large measles outbreaks may not be an effective indicator of under-resourced health systems in LMICs. Given these findings, we cannot make any recommendations on the aspects of the health system that warrant specific attention due to their indication of vulnerability to measles outbreaks. We recommend further research to investigate the association between under-resourced health systems and the presence or absence of a measles outbreak. We also recommend that future research explore the possibility of this association at a sub-national level (including outbreaks that occurred on a local scale) and include a range of health system indicators that represent all six WHO building blocks of health systems [28].

Measles outbreaks remain a public health issue worldwide, with an increasing number of large and disruptive measles outbreaks in the last year, even 60 years after the introduction of a highly effective measles vaccine [2,3,6]. The ability to identify health systems that are particularly vulnerable to public health disasters, such as measles outbreaks, would greatly improve the effective allocation of health resources, enhancing the accessibility and availability of care for many groups.

## 5. Conclusions

This analysis found no association between under-resourced health systems and the likelihood of a large measles outbreak in LMICs. These results highlight the complex and interconnected nature of various components of health systems and indicate that measles outbreaks alone may not serve as an effective tracer of weaknesses in the health system. Future research should investigate whether under-resourced health systems are more likely to report measles outbreaks, to determine if past measles outbreaks can be used to identify priorities in health planning and mitigate future outbreaks.

## Figures and Tables

**Figure 1 vaccines-13-00367-f001:**
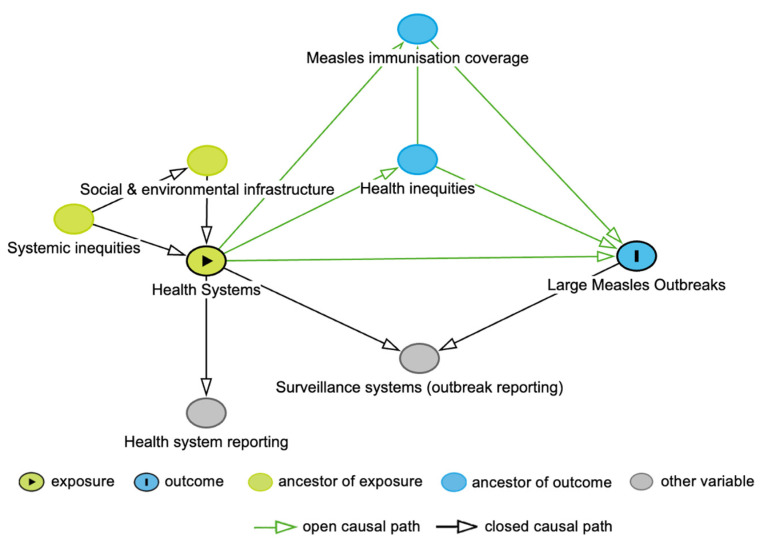
Directed Acyclic Graph for the association between health systems and large measles outbreaks.

**Figure 2 vaccines-13-00367-f002:**
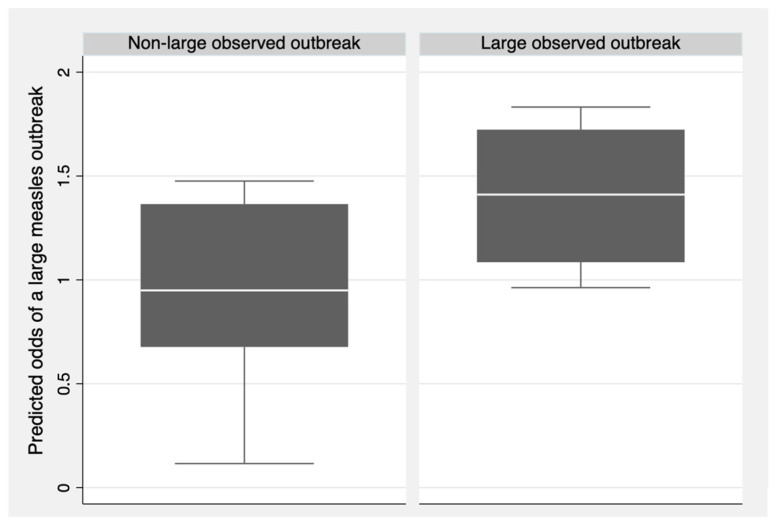
Predicted odds of large measles outbreaks compared to observed measles outbreak size for the testing set (*N* = 11).

**Table 1 vaccines-13-00367-t001:** Health system indicators stratified by large and non-large measles outbreaks (*N* = 53).

Health System Indicator (Median and IQR)	Large Measles Outbreaks (*N* = 28)	Non-Large Measles Outbreaks (*N* = 25)
Proportion of births delivered in a health facility (%)	74.90 (35.40, 98.20)	88.55 (57.40, 98.90)
Number of nurses and midwives (per 10,000 population)	10.96 (3.98, 45.52)	18.33 (7.97, 29.81)
Health expenditure (per capita in USD)	18.80 (7.14, 116.45)	97.16 (13.94, 260.50)

**Table 2 vaccines-13-00367-t002:** Predictors of large measles outbreaks in LMICs (multivariable regression).

Input Variable	Adjusted Odds Ratio	95% Confidence Intervals	*p*-Value
Proportion of births delivered in a health facility (%)	0.99	(0.96, 1.03)	0.74
Number of nurses and midwives (per 10,000 population)	1.03	(0.99, 1.08)	0.14
Health expenditure (per capita in USD)	0.99	(0.98, 1.00)	0.11

## Data Availability

The raw data supporting the conclusions of this article will be made available by the authors on request.

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
