# Peer review of "Using Measles Outbreaks to Identify Under-Resourced Health Systems in Low- and Middle-Income Countries: A Predictive Model"

_vaccines, 2025, doi:10.3390/vaccines13040367_

Round 1

Reviewer 1 Report

Comments and Suggestions for Authors

The authors undertook an investigation into a relationship between measles outbreaks and socioeconomic indicators in low and middle income countries. The investigation needs to document the process of statistical planning. Which countries were included? What was the source of information for each. What was the definition of Low and middle income come ?The authors need to analyse the percentage of immigrants from low income countries for each country included (refugees, asylum seekers) in relation to measles outbreaks.

Author Response

Comment 1: The investigation needs to document the process of statistical planning. Which countries were included? What was the source of information for each. What was the definition of Low and middle income?

Response 1: Thank you for bringing these omissions to our attention. The definition of low- and middle-income countries used in our analysis has now been to the Methods section at lines 221-223: “A country was considered an LMIC if it was listed by the World Bank as a low, lower-middle, or upper-middle income country in the calendar year that the outbreak began.” We have also uploaded the supplementary data file, which lists the measles outbreaks and the countries where they occurred.

Comment 2: The authors need to analyse the percentage of immigrants from low income countries for each country included (refugees, asylum seekers) in relation to measles outbreaks.

Response 2: Thank you for your thoughtful comment. We agree that measles outbreaks pose a considerable burden on migrant populations in LMICs, particularly in refugee camps and conflict-affected settings. However, incorporating a measure for these populations presents challenges, as they may have interacted with various health systems throughout their migration and often experience limited healthcare access while living in refugee/displaced people camps. Consequently, accurately accounting for the role of migrant populations in our analysis would require health system indicators specific to these populations. While measures of healthcare access and health outcomes have been developed for migrant populations, indicators of the broader health system are not yet established in the literature. Comments addressing these challenges as well as the burden of measles in migrant communities have been added to the Discussion in lines 459-464: “The presence of conflict may disrupt health services and access to healthcare, and measles outbreaks are frequently reported in refugee camps and conflict-affected populations. Relating health systems to displaced populations presents challenges due to migration and limited healthcare access. While not specifically examined in this study, further research is warranted to determine appropriate health system indicators for displaced populations.”

Reviewer 2 Report

Comments and Suggestions for Authors

Dear Authors,

I think the article is well done, below are some suggestions:

-I suggest not to use the acronym in the title of Charapter 3.1.

-I suggest revising the aim of the study, especially lines 123-126 should be put in first. Or at any rate, I suggest putting the aim as the last part of the introduction.

-For the discussion I suggest the following article to improve the sentence in line 370-372 :

Guarducci G, Porchia BR, Lorenzini C, Nante N. Overview of case definitions and contact tracing indications in the 2022 monkeypox outbreak. Infez Med. 2023 Mar 1;31(1):13-19. doi: 10.53854/liim-3101-3. 

-The conclusion should be improved (in particular line 422-424). 

Kind regards

Author Response

Comment 1: I suggest not to use the acronym in the title of Chapter 3.1.

Response 1: Thank you for pointing this out, it has now been amended (line 334).

Comment 2: I suggest revising the aim of the study, especially lines 123-126 should be put in first. Or at any rate, I suggest putting the aim as the last part of the introduction.

Response 2: Thank you for this suggestion. We have removed the lines detailing the study implications, as these are explained elsewhere. The last paragraph of the Introduction now only includes the revised aim, at lines 150-151: “This study aims to determine whether under-resourced health systems are associated with increased odds of large measles outbreaks in LMICs.”

Comment 3: For the discussion I suggest the following article to improve the sentence in line 370-372 : Guarducci G, Porchia BR, Lorenzini C, Nante N. Overview of case definitions and contact tracing indications in the 2022 monkeypox outbreak. Infez Med. 2023 Mar 1;31(1):13-19. doi: 10.53854/liim-3101-3. 

Response 3: Thank you for sharing this interesting article. We have included it as a reference at line 521.

Comment 4: The conclusion should be improved (in particular line 422-424). 

Response 4: Thank you for bringing this to our attention. We have now clarified the results and implications of the study at lines 553-559 to more directly relate to the aim: “This analysis found no association between under-resourced health systems and the likelihood of a large measles outbreak in LMICs. These results highlight the complex and interconnected nature of various components of health systems and indicate that measles outbreaks alone may not be an effective tracer of weaknesses in the health system. Future research should investigate whether under-resourced health systems are more likely to report measles outbreaks, to determine if past measles outbreaks can be used to identify priorities in health planning and mitigate future outbreaks.”

Reviewer 3 Report

Comments and Suggestions for Authors

General comment:

The manuscript “Using measles outbreaks to identify under-resourced health systems in low-and middle-income countries: a predictive model” has been reviewed. The study deals with a topic of interest and has a novel approach, but the characteristics of the surveillance system should also be taken into account because the capacity to detect early outbreaks is an essential component to control outbreaks and, consequently, to reduce the number of cases involved. In addition, there are a lot of repetitions in the introduction and discussion sections and so inaccuracies that affect the conclusions (see specific comments).

Specific comments:

Abstract:

“measles is a vaccine-preventable infectious disease” should be changed to “measles is a vaccine-preventable disease”.

Introduction:

Line 38: “measles is a vaccine-preventable infectious disease” should be changed to “measles is a vaccine-preventable disease”.

Lines 45-47: The large and disruptive measles outbreaks are defined according to reference [5] as “those with an incidence of at least 20 cases per million populations over a period of 12 months”. However, in the Material and Methods section authors define a large outbreak as having an incidence equal to or greater than 100 cases per million population over a 12-month period and in addition, they conduct a sensitivity analysis for different definitions: incidence greater than or equal to 50 cases, 150 cases and 200 cases per million population over a course of 12 months. However, if authors consider that having an incidence equal or greater than 100 cases per million population is better than that of at least 20 cases per million population that is supported by reference [5] a justification is needed.

Lines 58-61, lines 69-73, lines 77-79 and 96-98: These sentences are very similar and should be unified to avoid repetitions.

Lines 92-93: “preventing and responding to outbreaks” implies mainly the surveillance system because capacity to detect early outbreaks is needed to give adequate response, but the surveillance system, although is included in figure 1, is not mentioned in the Introduction and neither in the other sections of the manuscript.

Line 99: The title of the reference [26] should be deleted because readers can see it looking at the mentioned reference

Line 105: The title of the reference [30] should be deleted because readers can see it looking at the mentioned reference

Lines 119-125: The aims of the study should be clearly stated: Is the aim of the study to determine whether under-resourced health systems are associated with an increased probability of large measles outbreaks in LMICs or is also an aim of the study to compare if large outbreaks provides more robust data than smaller outbreaks?

Lines 123-126: The sentence “Understanding the association between under-resourced health systems and large outbreaks may help public health professionals to identify those health systems which are more at risk of a large measles outbreak and apply appropriate interventions” is not an objective of the study and the concept has also explained previously.

Material and Methods:

Line 144: The title of the reference [34] should be deleted because readers can see it looking at the mentioned reference

Lines 164-172: The characteristics of the surveillance system should also be taken into account because the capacity to detect early outbreaks is an essential component to control outbreaks and, consequently, to reduce the number of cases involved.

Lines 193-195: the rationale for the definition of large outbreak should be included.

Line 214: The title of this subsection could be “2.4.1. Regression model fitting and testing” and therefore, line 222 could be deleted.

Results:

Lines 248-249: The authors state “We also considered the measurement of the health system and large measles outbreaks through both health system reporting and surveillance systems”. However, no mention about the components of the surveillance system that have been considered.

Line 261: “specified time period” should be changed to “study period”

Table 1: the title is not precise: 53 is the number of outbreaks, but it seems to be related to the input variables. In addition to the three indicators included, an indicator of the surveillance system should be included

Figure 2: It is not clear: the ordinate axis is “predicted odds of a large measles outbreak” but in the abscissa axis there are both: “Non-large outbreak” and “Large outbreak”

Discussion:

Lines 337-340: Authors state that the selection of only three indicators id justified. However, as mentioned previously, in my opinion the components of the surveillance system should also be included. In fact, in lines 375-377 it is stated “The size of existing measles outbreak depends on both prevention and control measures such as reactive vaccination campaigns.

Table A1. The title should be “Sensitivity analysis results for different large measles outbreak definition”

“≥50 cases” should be changed by “≥50 cases per million”

“≥150 cases” should be changed by “≥150 cases per million”

“≥200 cases” should be changed by “≥200 cases per million”

Taking into account that in reference [5] it is stated that “large measles outbreaks are those with an incidence of at least 20 cases per million population over a period of 12 months”, why the category 20-49 cases per million has not be considered in the sensitivity analysis?

References:

In references 2, 13, 14, 20, 23, 27, 30, 39, 52, 54, 56 and 57 the first letter of all the words of the title are capitalized but in the other references only the first word of the title is capitalized and in the other words the first letter is small.

In references 6, 12, 26 and 35 the title of the document is in italics and should be in roman letters.

Author Response

Comment 1: Abstract: “measles is a vaccine-preventable infectious disease” should be changed to “measles is a vaccine-preventable disease”. 

Response 1: Thank you for pointing this out, it has now been amended (line 12).

Comment 2: Line 38: “measles is a vaccine-preventable infectious disease” should be changed to “measles is a vaccine-preventable disease”.

Response 2:  Thank you, this has now been changed (line 38).

Comment 3: Lines 45-47: The large and disruptive measles outbreaks are defined according to reference [5] as “those with an incidence of at least 20 cases per million populations over a period of 12 months”. However, in the Material and Methods section authors define a large outbreak as having an incidence equal to or greater than 100 cases per million population over a 12-month period and in addition, they conduct a sensitivity analysis for different definitions: incidence greater than or equal to 50 cases, 150 cases and 200 cases per million population over a course of 12 months. However, if authors consider that having an incidence equal or greater than 100 cases per million population is better than that of at least 20 cases per million population that is supported by reference [5] a justification is needed.

Response 3: Thank you for this important consideration. In the Methods section (lines 275-283), we have added a justification for our definition of a large measles outbreak in the context of the WHO definition given in reference [5]: “The WHO defines a large and disruptive measles outbreak as having an incidence of at least 20 laboratory-confirmed or clinically-compatible cases per million population over 12 months. Investigations of past measles outbreaks in LMICs have found that between 15% and 37% of suspected cases are confirmed as positive by a serological laboratory assay. As our analysis includes suspected, laboratory-confirmed and epidemiologically-linked cases, we defined a large outbreak as having an incidence equal to or greater than 100 cases per million population over a 12-month period, as we expect approximately 20 of these cases to be laboratory-confirmed.”

Comment 4: Lines 58-61, lines 69-73, lines 77-79 and 96-98: These sentences are very similar and should be unified to avoid repetitions.

Response 4: Thank you for bringing this to our attention. We have removed any repeated information and condensed these sentences into lines 69-75: “In LMICs, challenges to achieving high, uniform vaccination coverage are exacerbated by a decreased uptake in routine childhood vaccines following the COVID-19 pandemic and the effect of previous parasite infections on vaccine-induced immune responses. In under-resourced settings, where health systems lack sufficient infrastructure to deliver effective vaccination programs and control outbreaks of the disease, measles can become a considerable public health burden.”

Comment 5: Lines 92-93: “preventing and responding to outbreaks” implies mainly the surveillance system because capacity to detect early outbreaks is needed to give adequate response, but the surveillance system, although is included in figure 1, is not mentioned in the Introduction and neither in the other sections of the manuscript.

Response 5: Thank you for this thoughtful insight. We agree that surveillance systems play a critical role in detecting and responding to measles outbreaks. In our study, surveillance systems are incorporated into our DAG (Figure 1) as they are both influenced by the broader health system and serve as a key source of outbreak reporting. However, including a measure of surveillance systems in our analysis would open the health system ® surveillance systems pathway and the large measles outbreaks ® surveillance systems pathway, introducing collider bias. This would conflate the direct association between health systems and measles outbreaks with the association transmitted through surveillance systems. Additionally, our study focusses on broader health system components as outlined in WHO’s Monitoring the Building Blocks of Health Systems, which collectively underpin the effectiveness of surveillance systems.

We have added comments on the importance of surveillance systems and our approach to characterising the heath system to the Introduction at lines 93-136: “Robust health systems, including strong surveillance systems, are crucial to preventing and responding to outbreaks of vaccine-preventable diseases. Accessible health services, a strong health workforce, and efficient communication within the health system encourage high immunisation coverage. Surveillance plays a critical role in early detection and response, therefore limiting the size of emerging outbreaks. The WHO describes the core components of the health system using six distinct building blocks: health service delivery, the health workforce, access to essential medicines, health information systems, health financing, and leadership & governance. The building blocks framework has been widely used in the health system strengthening literature, while also serving as a basis for health care evaluation toolkits. In this analysis, we examine these fundamental aspects of the health system to unveil the underlying gaps that hinder outbreak prevention and response.”

Comment 6: Line 99: The title of the reference [26] should be deleted because readers can see it looking at the mentioned reference

Response 6: Thank you, this title has now been deleted (line 130).

Comment 7: Line 105: The title of the reference [30] should be deleted because readers can see it looking at the mentioned reference

Response 7: Thank you, this title has now been deleted (line 134).

Comment 8: Lines 119-125: The aims of the study should be clearly stated: Is the aim of the study to determine whether under-resourced health systems are associated with an increased probability of large measles outbreaks in LMICs or is also an aim of the study to compare if large outbreaks provides more robust data than smaller outbreaks?

Response 8: Thank you for this suggestion. The study aim has now been clarified at lines 150-151: “This study aims to determine whether under-resourced health systems are associated with increased odds of large measles outbreaks in LMICs.”

Comment 9: Lines 123-126: The sentence “Understanding the association between under-resourced health systems and large outbreaks may help public health professionals to identify those health systems which are more at risk of a large measles outbreak and apply appropriate interventions” is not an objective of the study and the concept has also explained previously.

Response 9: Thank you for pointing this out, we agree that there is superfluous detail in the last paragraph of the Introduction, so we have removed the lines detailing the study implications, as these are explained elsewhere.

Comment 10: Line 144: The title of the reference [34] should be deleted because readers can see it looking at the mentioned reference

Response 10: Thank you, this title has now been deleted (line 221).

Comment 11: Lines 164-172: The characteristics of the surveillance system should also be taken into account because the capacity to detect early outbreaks is an essential component to control outbreaks and, consequently, to reduce the number of cases involved.

Response 11: Thank you for this suggestion. As discussed in our response to Comment 5, we have addressed the importance of surveillance systems and have provided an explanation of our approach in lines 93-136.

Comment 12: Lines 193-195: the rationale for the definition of large outbreak should be included.

Response 12: Thank you for this important consideration. In the methods section (lines 275-283), we have added a justification for our definition of a large measles outbreak in the context of the WHO definition given in reference [5].

Comment 13: Line 214: The title of this subsection could be “2.4.1. Regression model fitting and testing” and therefore, line 222 could be deleted.

Response 13: Thank you, this has now been updated (line 300).

Comment 14: Lines 248-249: The authors state “We also considered the measurement of the health system and large measles outbreaks through both health system reporting and surveillance systems”. However, no mention about the components of the surveillance system that have been considered.

Response 14: Thank you for this consideration. As discussed in our response to Comment 5, we have addressed the importance of surveillance systems and have provided an explanation of our approach in lines 93-136.

Comment 15: Line 261: “specified time period” should be changed to “study period”

Response 15: Thank you, this has now been amended (line 360).

Comment 16: Table 1: the title is not precise: 53 is the number of outbreaks, but it seems to be related to the input variables. In addition to the three indicators included, an indicator of the surveillance system should be included

Response 16: Thank you for pointing out this inaccuracy. We have updated the title to “Health system indicators stratified by large and non-large measles outbreaks (N = 53)” at lines 368-369. We have chosen not to include surveillance systems as an input variable as discussed in our response to Comment 5.

Comment 17: Figure 2: It is not clear: the ordinate axis is “predicted odds of a large measles outbreak” but in the abscissa axis there are both: “Non-large outbreak” and “Large outbreak”

Response 17: Thank you for bringing our attention to this error. On the abscissa axis, the observed outcomes are displayed, so we have updated these axis labels to “Non-large observed outbreak” and “Large observed outbreak”, to distinguish between these observed outcomes and the predicted odds displayed on the ordinate axis.

Comment 18: Lines 337-340: Authors state that the selection of only three indicators id justified. However, as mentioned previously, in my opinion the components of the surveillance system should also be included. In fact, in lines 375-377 it is stated “The size of existing measles outbreak depends on both prevention and control measures such as reactive vaccination campaigns.

Response 18: Thank you for this comment. As discussed in our response to Comment 5, we have addressed the importance of surveillance systems and have provided an explanation of our approach in lines 93-136. We have provided additional clarification in the Discussion at lines 448-452: “Such a relationship between the building blocks also suggests that including additional building block indicators or a surveillance system measure in our model may have led to collinearity, so selection of only three indicators is justified. The reliance of surveillance systems on the health system, coupled with their role in measles outbreak reporting, may have introduced collider bias had they been incorporated into our model.”

Comment 19: Table A1. The title should be “Sensitivity analysis results for different large measles outbreak definition”

“≥50 cases” should be changed by “≥50 cases per million”

“≥150 cases” should be changed by “≥150 cases per million”

“≥200 cases” should be changed by “≥200 cases per million”

Response 19: Thank you, we have made these changes in Table A1.

Comment 20: Taking into account that in reference [5] it is stated that “large measles outbreaks are those with an incidence of at least 20 cases per million population over a period of 12 months”, why the category 20-49 cases per million has not be considered in the sensitivity analysis?

Response 20: Thank you for this important consideration. We have now added 20 cases per million population as a category in the sensitivity analysis, and the results are recorded in Table A1. We have also updated the Methods section to reflect this additional category (line 286).

Comment 21: In references 2, 13, 14, 20, 23, 27, 30, 39, 52, 54, 56 and 57 the first letter of all the words of the title are capitalized but in the other references only the first word of the title is capitalized and in the other words the first letter is small.

Response 21: Thank you, we have now updated the references so that the first letter of each word in the titles is capitalised.

Comment 22: In references 6, 12, 26 and 35 the title of the document is in italics and should be in roman letters.

Response 22: Thank you, this has now been amended.

Round 2

Reviewer 1 Report

Comments and Suggestions for Authors

The authors have not as requested documented statistical planning. Burden of countries by immigrants needs to be documented and analysed.

Reviewer 3 Report

Comments and Suggestions for Authors

The revised version of the manuscript entitled “Using measles outbreaks to identify under-resourced health systems in low-and middle-income countries: a predictive model” has been reviewed again. Despite many of my comments to the previous version have been taken into account by authors in the revised version, there are two comments on crucial aspects of the study that have not been considered.

One aspect is the definition of large outbreak that use the authors. In my opinion, this definition does not seem appropriate and compromises seriously the results. Authors state that “WHO defines a large and disruptive measles outbreak as having an incidence of at least 20 laboratory-confirmed or clinically-compatible cases per million populations over 12 months [5, 54, 55]”. This definition is supported by reference [5], being unnecessary reference [54] and reference [55].

According to WHO definition [WHO. Measles outbreak toolbox. https://www.who.int/emergencies/outbreak-toolkit/disease-outbreak-toolboxes/measles-outbreak-toolbox] a clinical case is any person in whom a clinician suspect measles infection or any person with fever and maculopapular rash and cough or coryza or conjuntivitis and in the WHO definition of large and disruptive measles outbreak not only laboratory-confirmed cases are included but also clinically-compatible cases. However, authors define a large outbreak as having an incidence equal to or greater than 100 cases per million population over a 12-month period because they expect approximately 20 of these cases to be laboratory-confirmed, without considering the clinical cases. In my opinion, analysis should be done using the WHO definition, that is to say, considering large outbreaks as having an incidence of at least 20-laboratory-confirmed or clinically compatible cases per million.

The second aspect is the implication of the surveillance system of the involved countries where the large measles outbreaks occurred. Authors have added in the revised version some comments about the importance of the surveillance systems but the surveillance system as a specific component of the health system has not been considered neither in methods nor in results.

Round 3

Reviewer 1 Report

Comments and Suggestions for Authors

The authors have adequately responded to my comments